# Regulatory Mechanisms of *Prg4* and *Gdf5* Expression in Articular Cartilage and Functions in Osteoarthritis

**DOI:** 10.3390/ijms23094672

**Published:** 2022-04-23

**Authors:** Yoshifumi Takahata, Hiromasa Hagino, Ayaka Kimura, Mitsuki Urushizaki, Shiori Yamamoto, Kanta Wakamori, Tomohiko Murakami, Kenji Hata, Riko Nishimura

**Affiliations:** Department of Molecular and Cellular Biochemistry, Osaka University Graduate School of Dentistry, 1-8 Yamadaoka Suita, Osaka 565-0871, Japan; h-hagino@dent.osaka-u.ac.jp (H.H.); u961909e@ecs.osaka-u.ac.jp (A.K.); u170699g@ecs.osaka-u.ac.jp (M.U.); shioriyamamotoimamura@dent.osaka-u.ac.jp (S.Y.); ohio.ohayou8.monash@gmail.com (K.W.); tmurakami@dent.osaka-u.ac.jp (T.M.); hata@dent.osaka-u.ac.jp (K.H.); rikonisi@dent.osaka-u.ac.jp (R.N.)

**Keywords:** osteoarthritis, *Prg4*, Transcription factor, Wnt, β-catenin, EGFR, TGF-β

## Abstract

Owing to the rapid aging of society, the numbers of patients with joint disease continue to increase. Accordingly, a large number of patients require appropriate treatment for osteoarthritis (OA), the most frequent bone and joint disease. Thought to be caused by the degeneration and destruction of articular cartilage following persistent and excessive mechanical stimulation of the joints, OA can significantly impair patient quality of life with symptoms such as knee pain, lower limb muscle weakness, or difficulty walking. Because articular cartilage has a low self-repair ability and an extremely low proliferative capacity, healing of damaged articular cartilage has not been achieved to date. The current pharmaceutical treatment of OA is limited to the slight alleviation of symptoms (e.g., local injection of hyaluronic acid or non-steroidal anti-inflammatory drugs); hence, the development of effective drugs and regenerative therapies for OA is highly desirable. This review article summarizes findings indicating that proteoglycan 4 (*Prg4*)/lubricin, which is specifically expressed in the superficial zone of articular cartilage and synovium, functions in a protective manner against OA, and covers the transcriptional regulation of *Prg4* in articular chondrocytes. We also focused on growth differentiation factor 5 (*Gdf5)*, which is specifically expressed on the surface layer of articular cartilage, particularly in the developmental stage, describing its regulatory mechanisms and functions in joint formation and OA pathogenesis. Because several genetic studies in humans and mice indicate the involvement of these genes in the maintenance of articular cartilage homeostasis and the presentation of OA, molecular targeting of *Prg4* and *Gdf5* is expected to provide new insights into the aetiology, pathogenesis, and potential treatment of OA.

## 1. Introduction

Articular cartilage is an important locomotor tissue with many functions such as wear resistance, gliding properties, friction reduction, and the provision of load-bearing surfaces [1]. Osteoarthritis (OA), the most common degenerative joint disease, which affects 300 million people worldwide [2,3]. Aging and genetic constitution are the main risk factors for OA, which often leads to stiffness and the loss of mobility in elderly individuals. The pathogenesis of OA involves the loss of aggrecan, proteoglycans, and other substances, resulting from an imbalance between the synthesis and degradation of the cartilage matrix. This loss causes morphological changes to collagen 10 α1 (Col10α1)-positive hypertrophic chondrocytes in the joint surface layer [4,5], as well as the formation of osteophytes and subchondral sclerosis, with the progression of OA [6]. Inflammatory cytokines interleukin 1 (IL-1) and tumor necrosis factor (TNF)-α, both of which are commonly increased in synovial fluid during OA, are thought to strongly induce expression of the catabolic enzymes matrix metallopeptidase 13 (Mmp13) and ADAM metallopeptidase with thrombospondin type 1 motif 4 and 5 (Adamts4/5) [7,8] to promote the progression of cartilage destruction.

Genetic studies performed in mice showed that knockout (KO) of *Mmp13* or *Adamts5* suppressed the progression of OA occurring spontaneously or induced by surgical destabilization of the medial meniscus (DMM) [9,10,11,12,13]. Adamts4 does not have the same predominant function as Adamts5 in mice but seems to be involved in joint destruction in humans [14,15]. Therefore, it is desirable to develop therapeutic drugs that selectively downregulate expression of these catabolic enzymes.

Proteoglycan 4 (*Prg4*), also known as lubricin, an extracellular matrix molecule essential for maintaining lubrication and homeostasis of the joint surface layer, has also been implicated in the development of OA [16]. The growth differentiation factor 5 (*Gdf5*) gene is important for joint formation, and there is a close relationship between the function of this gene and the development of OA. *Gdf5* showed potent stimulation of extracellular matrix accumulation, type II collagen, and aggrecan expression and also decreased the expression of catabolic factors, namely MMP13 and ADAMTS5, in cultures of human OA chondrocytes [17].

Accordingly, understanding the precise roles of intracellular signaling and transcriptional regulation involved in controlling the expression of *Prg4* and *Gdf5* in articular cartilage (e.g., by examining aspects of cartilage matrix synthesis), has the potential to prevent OA onset and/or progression, or provide remittance after its onset. Therefore, in this review, we aimed to comprehensively summarize the functional roles of *Prg4* (Table 1) and *Gdf5* (Table 2) in relation to OA, and describe the molecular mechanisms involved in regulating their transcription.

## 2. *Prg4* Is Essential for Maintaining the Health and Homeostasis of Cartilage

*Prg4*, a proteoglycan specifically expressed in the articular surface layer and synovium, has lubricating properties that reduce friction in joints. In humans, genomic deletions or mutations of *PRG4* have been implicated in the development of camptodactyly-arthropathy-coxa vara-pericarditis syndrome [18], which is associated with congenital strabismus and childhood-onset non-inflammatory arthropathy; notably, no specific joint abnormalities are observed immediately after birth [19].

An analysis of mouse embryos showed that *Prg4* expression commences at embryonic day 15.5. Ref. [20] remains in the joint surface until at least postnatal day 30, and then spreads toward the deeper layers of articular cartilage with aging [21]. Passive mechanical stress during rehabilitation induced *Prg4* expression in bovine knee joints and contributed to the maintenance of joint health [22].

Several reports have linked Prg4 depletion in the knee joint with OA development. First, the physical reduction of friction at the joint boundary was observed in *Prg4* KO mice [23]. Additionally, *Prg4* KO mice showed changes in cell morphology of the joint surface layer at 2 weeks of age and progressive destruction of the joint surface layer at 16 weeks of age [24]. Moreover, apoptosis-related markers such as caspase 3 were elevated in the knee joint tissue of *Prg4* KO mice, indicating a correlation between the maintenance of joint tissue lubrication and the apoptosis of chondrocytes [25]. In contrast, transgenic mice overexpressing *Prg4* under control of the cartilage-specific *Col2**α1* gene promoter exhibited an attenuated degree of joint destruction accompanying age-related OA and post-traumatic OA compared with wild-type mice [16,26]. Notably, it is feasible to overexpress *Prg4* by directly injecting the hepatitis D virus or adeno-associated virus 2 into the joint cavity, whereby it functions to protect against OA development [16].

Collectively, these findings suggest that Prg4 expressed in the superficial zone (SFZ) cells of articular cartilage is essential for joint homeostasis. Moreover, investigating the molecular mechanisms that regulate *Prg4* expression will contribute to our understanding of the physiological and pathological properties of articular cartilage, as well as the development of novel and effective therapies for OA.

## 3. Transforming Growth Factor (TGF)-β Signaling Regulates *Prg4* Expression

The binding of TGF-β to type I (activin-linked kinase [ALK]5) and type II (TGFβR2) receptors, both of which belong to the serine/threonine kinase receptor family, promotes their kinase activity and the phosphorylation of Smad2 and Smad3, the downstream effector proteins of TGF-β (Figure 1). Phosphorylated Smad2 or Smad3 subsequently associates with Smad4, a co-regulatory Smad protein, and is translocated from the cytoplasm to the nucleus; consequently, the complex activates the transcription of downstream target genes (Figure 1). TGF-β signaling induces *Prg4* expression [27,28,29] through a Smad3-mediated signaling pathway [30] (Figure 1). Moreover, a cartilage explant experiment showed that TGF-β promotes *Prg4* secretion [31]. Mice overexpressing dominant-negative TGF-β receptor 2 (TGFβR2) exhibited reduced *Prg4* expression in the joint surface layer, thinner joints, and progressive cartilage fibrillation [26]. These observations indicate that TGF-β signaling is involved in the stimulation of *Prg4* expression in articular chondrocytes.

Interestingly, autologous platelet-rich fibrin, a second-generation platelet product, activates TGF-β signaling, phosphorylates Smad3, and enhances *Prg4* expression [32]. Thus, it can improve articular cartilage repair and is becoming established as an effective pharmacotherapy for OA [33].

## 4. Importance of Wnt Signaling in Articular Cartilage

Several studies indicate that Wnt proteins, which play important roles in the proliferation, differentiation, and development of numerous types of cells and tissues, are involved in the homeostasis of articular cartilage. Wnt proteins bind to the seven-transmembrane receptors of the Frizzled family and cooperate with the co-receptors low-density lipoprotein receptor-related proteins 5 and 6 (LRP5 and LRP6) for intracellular signaling [32] (Figure 2). Wnt is a well-conserved family of secreted proteins present in various tissues [34,35]. There are three major Wnt signaling pathways: the canonical Wnt/β-catenin pathway, the Wnt/Ca^2+^ pathway, and the non-canonical cell polarity pathway. In canonical Wnt signaling, when Wnt is not bound to its receptor complex, β-catenin is phosphorylated by serine/threonine kinases, casein kinase (CK) 1, and glycogen synthase kinase 3β (Gsk3β). Phosphorylated β-catenin becomes a target of β-transducin repeat-containing protein (a component of the E3 ubiquitin ligase complex) and is rapidly degraded by the proteasome. After the binding of Wnt to its receptor complex, the Frizzled/LRP co-receptor complex binds axin (an upstream protein of GSK3β and β-catenin) to block proteasomal degradation, resulting in β-catenin stabilization [36] (Figure 2). Subsequently, β-catenin translocates from the cytoplasm into the nucleus, where it replaces Groucho bound to transcription factor 7 (TCF)/lymphoid enhancer binding factor (LEF) and induces the transcription of Wnt target genes.

Several studies have implicated the canonical Wnt signaling pathway in the pathogenesis of OA. Recently, the canonical Wnt pathway was proposed as both a risk factor for OA and an important signal for determining the physiological characteristics of the articular surface layer through the regulation of *Prg4* expression [37]. A conditional transgenic mouse model in which constitutively active β-catenin is overexpressed in articular cartilage showed a loss of SFZ cells in articular cartilage and severe OA-like symptoms with aging [38]. The activation of β-catenin by Wnt3a stimulation increased the expression of *Mmp3*, *Mmp13*, *Adamts4,* and *Adamts5*, resulting in the degradation of cartilage matrices [39] (Figure 2).

A comprehensive microarray analysis of injured human joint tissues cultured ex vivo to identify genes whose expression was altered by injury showed that *Wnt16* expression was markedly increased [40]. Additionally, β-catenin expression levels were increased in human OA samples [38]. Wnt signaling was activated in mice deficient for Frizzled-related protein (*Frzb*), a secreted WNT antagonist, while *Frzb* KO mice showed more severe joint destruction than wild-type mice in collagen-induced and surgically treated OA models [41,42]. Thus, several gain-of-function studies have shown that the activation of Wnt/β-catenin signaling promotes the progression of joint destruction and OA-like symptoms.

Loss-of-function studies of Wnt/β-catenin showed a different aspect of Wnt signaling in articular cartilage. Mice with a transgenic expression of the inhibitor of β-catenin and TCF peptide, which have disrupted β-catenin/TCF binding, exhibited suppressed Wnt signaling [43], increased cartilage apoptosis, and an accelerated progression of joint destruction [44]. Notably, mice with a conditional KO of β-catenin controlled by the *Col2a1* promoter lost the articular surface region where *Prg4* is highly expressed, whereas mice with conditionally active β-catenin controlled by the *Col11a2* promoter exhibited increased numbers of BrdU-labeled SFZ cells and *Prg4*-positive cells in the articular surface region [45]. Furthermore, *Prg4* expression was increased when SFZ cells were exposed to Wnt3a [45].

To understand the physiological role of cells specifically localized in the articular surface region, Xuan et al., recently analyzed *Prg4*-CreERT2; *Ctnnb1*^flox/flox^ conditional KO mice and *Prg4*-CreERT2; and *Ctnnb1*-ex3^flox/+^ conditionally active mice. Their results showed that conditional β-catenin KO mice exhibited reduced *Prg4* expression in SFZ cells, and that β-catenin stabilization induced *Prg4* expression [46].

During the progression of OA, inflammatory cytokines such as IL-1β, IL-6, and nerve growth factor are increased in the knee joint [47]. Sclerostin, a Wnt antagonist, is induced by IL-1 and TNF-α and upregulated in the DMM-induced post-traumatic OA mouse model [48]. Recombinant sclerostin downregulated the gene expression of catabolic factors such as Adamts5 and protected against cartilage destruction in an OA mouse model [48]. Additionally, compared with wild-type mice, sclerostin transgenic mice were protected from joint destruction in a post-traumatic OA model established by injury of the anterior cruciate ligament [49]. Additionally, Wnt/β-catenin signaling has been reported to be essential for joint formation during skeletal development [50]. Therefore, these findings suggest that appropriate levels of Wnt/β-catenin signals are required to maintain cartilage homeostasis through induction of *Prg4* expression in adult articular joints after skeletal development. It is likely that appropriate control of Wnt/β-catenin signaling would be an effective approach to prevent the onset of OA.

## 5. EGFR Signaling Regulates *Prg4*

Epidermal growth factor receptor (EGFR) mediates important signals for tissue homeostasis and cell proliferation of various tissues. There are several ligands for EGFR, including EGF, TGF-α, amphireglin, eriregulin, and betacellulin [51]. TGF-α activates various signaling pathways such as Ras homolog family member A/Rho kinase, mitogen-activated protein kinase (MAPK)/extracellular signal-regulated kinase (ERK), phosphatidylinositol 3-kinase (PI3K), and p38 MAPK, and promotes the matrix degradation of articular cartilage [52]. EGFR signaling plays important roles in endochondral ossification, chondrocyte differentiation, and growth plate formation [53].

EGFR signaling was recently implicated in the regulation of *Prg4* expression in articular cartilage tissue, which may have a protective function against joint destruction [54,55,56]. Four weeks after DMM, EGFR phosphorylation was markedly reduced in SFZ cells of articular cartilage compared with a sham-operation control group. Similarly, EGFR phosphorylation was reduced in the knee joints of patients with early OA compared with non-OA knee joints [54].

Compared with control mice, OA progression was accelerated in both kinase-dead dominant negative EGFR (*EGFR^wa5/+^*) heterozygous mice and mice treated with gefinitib, an effective EGFR kinase inhibitor [57]. Cartilage and mesenchymal stem cell-specific KO mice conditional for mitogen inducible gene 6 (*Mig6*), an ErbB receptor feedback inhibitor placed under the control of the *Col2a1* and *Prx1* promoters, exhibited activated EGFR signaling and increased knee-joint thickness [58,59]. In contrast, mice with a joint-specific overexpression of *Mig6* showed no abnormalities in articular cartilage development but had decreased *Prg4* expression in articular cartilage and progressive joint destruction with aging [60]. Mice overexpressing heparin-binding EGF-like growth factor, an EGFR ligand placed under the control of the aggrecan gene promoter, exhibited highly activated EGFR signaling and attenuated OA progression [55]. Because the downstream signaling molecules of EGFR involved in the induction of *Prg4* expression remain elusive, careful and precise investigations are needed in the future.

## 6. *Prg4* Regulation by Transcription Factors

Numerous studies have revealed important transcription factors for the regulation of *Prg4* expression. Forkhead box class O (FOXO) transcription factors are downstream targets of the protein kinase Akt. The PI3K-Akt-FOXO signaling pathway has multiple cellular functions, including cell proliferation and survival [61,62]. The expression of FOxO1 and FOxO3 is reportedly decreased in the articular cartilage of patients with OA, while the expression of FoxO1 and FoxO3 is decreased in aged mice and DMM mice [63]. Meniscus damage is a well-known and major factor in the development of OA. FoxO1 and FoxO3 were shown to play important roles in meniscus development and homeostasis during aging and OA [64]. The ectopic expression of FoxO1 increased *Prg4* expression and synergistically upregulated *Prg4* expression with TGF-β. Cartilage-specific conditional FoxO1, FoxO3, and FoxO4 triple KO mice induced morphological changes in joint surface cells and spontaneous joint destruction [65]. These findings illustrate important roles of FOXO family members in joint homeostasis and protection from OA (Figure 3). 

Several studies investigated the critical role of the nuclear factor of activated T cells (NFATc) transcription factors in articular cartilage and *Prg4* expression (Figure 3). Immunohistological studies showed that *Nfatc1* is strongly expressed in the superficial region of articular cartilage, but joint-specific conditional *Nfatc1* KO mice did not exhibit affected chondrocytes [66]. In contrast, *Nfatc2* KO mice exhibit OA-like symptoms because of an imbalance between cartilage matrix degradation and synthesis [67]. Because *Nfatc1* and *Nfatc2* are in the same family and functionally compensate for one another, mice with a cartilage-specific deletion of *Nfatc1* and a deficiency in *Nfatc2* (*Nfatc1^col2^*; *Nfatc2^−/−^*) were generated and analyzed. These mice exhibited a reduced expression of *Prg4* in joint tissues that was accompanied by spontaneous and severe OA with 100% penetrance.

Mechanical exercise reportedly induces phosphorylation of the DNA-binding transcriptional regulator CreB (CREB) and increases *Prg4* expression [68]. In a more recent study, an RNA sequencing analysis of the superficial and deep layer cells of articular cartilage identified *CREB5* as being specifically expressed in the superficial layer [69]. Moreover, overexpression of *Creb5* strongly induced *Prg4* expression by functioning with the TGF-β and EGFR signaling pathways [69] (Figure 3). Because these in vitro findings suggest a key role of CREB5 in regulating *Prg4* expression, the relationship between CREB5 and *Prg4* expression awaits appropriate in vivo analyses.

## 7. GDF5 Signaling

GDF5 is a bone morphogenetic protein (BMP) also known as BMP-14 and CDMP-1 that belongs to the TGF-β superfamily of signaling proteins, which have important roles in skeletal formation and development.

The Bmp/Gdf family receptor is a heterotetrameric complex comprising the type 2 serine/threonine kinase receptors BMPR2, ACVR2a, and ACVR2B, and the type 1 receptors ALK2, BMPR1A (ALK3), and BMPR1B (ALK6). The strongest BMP inducers of osteogenic and chondrogenic differentiation, Bmp2 and Bmp4 form complex structures [70]. Although Bmp2 and Bmp4 have a similar affinity for BMPR1A and BMPR1B type 1 receptors, GDF5 was shown to have a stronger affinity for BMPR1B than for BMPR1A [71]. Furthermore, GDF5 mutants with a weakened ability to form complexes with BMPR1A and BMPR2 showed a reduced activity for osteochondral differentiation, indicating that BMPR1A is required for osteochondral differentiation. In contrast, a GDF5 mutant with a stronger BMPR1B and BMPR2 complex formation had an inhibitory effect on chondrocyte hypertrophy. Stronger BMPR1B signaling relative to BMPR1A signaling may suppress chondrogenic differentiation and contribute to stability during joint formation. When the BMP/GDF family binds to a receptor complex, the type I receptor (that has been activated by a type II receptor) phosphorylates the intracellular BMP signaling mediators Smad1, 5, and 8 at their C-terminal motifs. Although BMP/Gdf5 appear to share a common Smad signal, the Smad-induced gene expression profiles were recently found to be different; thus, more careful analysis of the differences in downstream signaling is needed. The transcriptional repressor GATA binding 1 (*Trps1*) gene, which has a similar expression pattern to *Gdf5*, has been reported to functionally interact with GDF5. Interestingly, the overexpression of *Gdf5* promoted *Trsp1* expression and the phosphorylation of p38 MAPK, and these effects were inhibited by a dominant-negative form of ALK6 [72]. These results indicate that Trsp1 acts downstream of GDF5, providing one explanation for the difference in functions between GDF5 and BMP.

## 8. *Gdf5* Functions in Cartilage Development

GDF5 also promotes osteogenesis [73] and is expressed in the developing limb at sites where the joint cavity is created [74]. *Gdf5* is strongly expressed until the fetal day 14 arthroplasty stage, but is downregulated later in development [20]. Null mutations in *Gdf5* prevent the formation of over 30% of the synovial joints in the extremities, result in partial fusion of certain joints, and alter the patterns of repetitive structures in the fingers, wrists, and ankles [74,75,76]. The addition of recombinant GDF5 to bone marrow-derived mesenchymal cells increased type X collagen expression, and promoted chondrogenic differentiation and hypertrophy of chondrocytes [77]. Subcutaneous implantation of collagen or collagen/hyaluronan containing GDF5 in rats, results in chondrogenesis followed by osteogenesis. The chondrogenic activity of GDF5 in vivo has also been demonstrated, as transgenic mice with a targeted expression of *GDF5* showed an increased chondrogenic capacity due to the thickening and hypertrophy of the cartilage component [78]. Patients with Grebe-type chondrodysplasia show a severe phenotype of shortened limbs and appendicular bone dysmorphogenesis. In the case of one affected family, a frameshift mutation in the *GDF5* gene was discovered that caused complete loss of GDF5 signaling, which is critical for the regulation of axial bone growth during skeletal development [79].

Developmental dysplasia of the hip (DDH) is the most common skeletal dysplasia. A genome-wide association study found that the genetic component of DDH is 55%, and that it is equally distributed on autosomal and X chromosomes, establishing a robust DDH gene-related locus in *Gdf5* [80]. Additionally, the promoter of the *GDF5* gene in cartilage samples from patients with DDH was hypermethylated compared with control samples from healthy adults, indicating that methylation might regulate *GDF5* expression and contribute to the pathogenesis of DDH [81]. Furthermore, non-methylated DNA in the 5’ untranslated region has been correlated with increased *Gdf5* expression in cell lines and joint tissues. In mice with a KO of the DNA methyltransferase 3A gene, methylation levels in the *Gdf5* gene promoter were reduced, while treatment with DNA methylation inhibitors increased *Gdf5* expression [82]. These findings also indicate that *Gdf5* expression is epigenetically regulated by DNA methylation [83].

In a bone defect rabbit model used to test the bone healing process, the administration of GDF5 showed delayed tissue mineralization compared with BMP2, causing cartilage tissue to be present in the defect [84]. Intra-articular administration of GDF5 promoted cartilage repair in a rat model of OA [85]. Although GDF5 has similar chondrogenic and ossifying abilities as Bmp2, its effects are weaker than those of Bmp2. Moreover, it has been shown that GDF5 plays an important role in maintaining joint function. Several gene expression and deletion studies strongly suggest that GDF5 (along with GDF6 and GDF7) has specialized functions in joint morphogenesis [74].

Single-cell RNA sequencing data generated from *Gdf5*-positive cells of embryonic mouse knee joints showed that doublecortin (*Dcx*) expression was enriched in a cluster of GDF5-positive cells. Dcx-positive cells were enriched in chondrocyte differentiation and cartilage development, and *Gdf5* expression was positively correlated with *Dcx* expression [86]. Lineage tracing of *Gdf5*-expressing cells identified a population of mesenchymal stem cells in the synovium of adult mice that is mostly negative for skeletal stem cell markers, including *Nestin*-green fluorescent protein, Leptin receptor, and Gremlin1. Following injury, the Yes1-associated transcriptional regulator (*Yap*) is upregulated in *Gdf5*-lineage cells, preventing synovial hyperplasia and reducing the contribution of these cells to cartilage repair [87]. To understand the time-dependent cell population characteristics of GDF5 in joint tissues, a lineage-tracing analysis using *Gdf5*-CreER;TdTomato knockin mice was also undertaken. Early specified *Gdf5*-positive cells were found to contribute to meniscus formation and ligament formation. Intermediately specified *Gdf5*-positive cells played a role in meniscus, ligament formation, and articular cartilage formation. Late specified *Gdf5*-positive cells contributed significantly to articular cartilage formation [88]. These findings clearly indicate an important role for *Gdf5* in the development of joints, including articular cartilage.

## 9. *Gdf5* Is Associated with the Development of OA

As described above, *GDF5* is a cartilage-specific growth factor, and studies aimed at elucidating its gene function have focused on joint formation and chondrocyte differentiation. Associations between *GDF5* gene mutations and many human skeletal dysplasias have been reported, including Hunter-Thompson syndrome [89], DuPan syndrome [90], Grebe chondrodysplasia [91], angel-shaped phalangoepiphyseal dysplasia [92], brachdactyly type C [93], and brachydactyly type A2 [94]. A loss-of-function analysis of *Gdf5* in mice showed a decrease in the number of fingers and shortened tubular bones. Reduced GDF5 levels may be a precipitating factor in the development of OA, and investigations of OA risk alleles indicate that intra-articular supplementation with human recombinant GDF5 has the potential to effectively alter the pathogenesis of OA [85].

In recent years, several reports have described associations between OA and various polymorphisms in multiple genes, including *Asporin*, *CALM1*, and *GDF5*. Abd Elazeem et al. found in a case-control analysis of human OA patients that a polymorphism in the upstream regulatory region of *GDF5* (+104T/C), a candidate gene for OA susceptibility, decreased the transcriptional activity of *GDF5* and was strongly associated with the development of OA [95]. Approximately 80% of the patients had susceptibility alleles for this polymorphism, indicating that people with this genetic polymorphism are around 1.8 times more likely to develop OA of the hip joint. Moreover, a random single-nucleotide polymorphism screening technique showed that a mutation in mouse *Gdf5* (W408R) had dominant-negative effects, with the W408R mutant mice exhibiting a brachypodism-like phenotype [96].

Using a *Gdf5*-LacZ reporter gene in DMM model mice with experimental OA, a study by Kania et al. evaluated the spatiotemporal activity of *Gdf5* regulatory elements following acute cartilage injury and repair. They found that *Gdf5* expression in articular cartilage was upregulated after DMM treatment and in human OA patients’ cartilages, as assessed by immunohistochemical and microarray analyses. The microRNA known as miR21-5p is a noncoding RNA involved in the regulation of *Gdf5* expression. Mice deficient in miR21-5p showed increased *Gdf5* expression, decreased expression of the catabolic factor MMP13, and an alleviation of OA progression [97].

Expression of *Gdf5* in specific joints requires the binding of transcription factors within multiple enhancer regions. An array of enhancers involved in the regulation of *Gdf5* expression are distributed over 100 kilobases of DNA that includes regions that lie both upstream and downstream of the *Gdf5* coding exons [98]. These enhancers also map to an orthologous large genomic region previously associated with common adult OA risk in humans. Hence, genome-wide association studies of *GDF5* would likely provide information that could deepen our understanding of OA pathogenesis.

*Gdf5* expression in damaged synovium increased during cartilage repair in mice and was inversely correlated with the expression of the transcription co-factor yes-associated protein (Yap). The overexpression of *Yap* suppressed *Gdf5* expression during cartilage formation in vitro. These results suggest that elevated *Gdf5* expression in articular cartilage and synovium is a general response to knee joint injury that depends on downstream regulatory sequences, and that Gdf5 may have important functions in tissue remodeling and repair after injury, supporting its association with OA risk.

The ETS transcription factor-related gene, *ERG*, was identified as a transcription factor that is regulated by a downstream signal of GDF5, and is involved in the formation, establishment, and maintenance of articular cartilage. A cross between conditional KO mice lacking *Erg* in the cartilage and *Gdf5*-Cre mice showed higher sensitivity to surgically induced OA and exhibited spontaneous OA-like symptoms with aging. Erg was upregulated in the affected areas of OA articular cartilage and was shown to be involved in the regulation of *Prg4* expression, with a function in reducing articular cartilage damage [99]. Therefore, Erg is likely to be a key transcription factor marker of articular cartilage.

## 10. Conclusions

The understanding of the molecular mechanisms underlying cartilage diseases such as OA and rheumatoid arthritis has progressed at a remarkable pace in the past decade. In particular, molecular, biological, genetic, and immunological studies have revealed the pathological involvements of specific intracellular signaling pathways and transcription factors in these diseases. Among these potential therapeutic targets, the establishment of anti-IL-6 and TNF-α antibodies, as well as Janus kinase (JAK) inhibitor therapies for rheumatoid arthritis, have been the most successful. However, identification of effective therapies for OA remains challenging and we are still far from achieving the ultimate goal of curing patients.

A recent clinical study demonstrated that the phenomenon of the meniscus being pushed out from its original location to the synovial side is associated with the development of OA of the knee, including changes in the length of osteophytes and the presence of hypertrophic chondrocytes in articular cartilage [100]. This phenomenon of osteophyte formation and hypertrophy due to the increased calcification of articular chondrocytes resembles the final step of endochondral osteogenesis, a mode of differentiation of growth plate chondrocytes. Therefore, research conducted globally on the pathogenesis of OA and the development of appropriate therapies have focused on elucidating the molecular mechanisms of endochondral osteogenesis. In particular, studies using KO mice for the transcription factors hypoxia-inducible factor 2α and CCAAT/enhancer binding protein β have contributed to understanding of the pathogenesis of OA [4,101,102]. However, articular chondrocytes have unique cell characteristics and gene expression profiles that differ greatly from those of growth plate chondrocytes. Furthermore, the differentiation patterns and lineages of superficial, middle, and deep zone cells remain poorly understood. The expression of *GDF5* and *Prg4* in joint tissues is important for the maintenance of articular cartilage homeostasis. The dissection of molecular mechanisms involved in regulating *Prg4* expression and understanding the downstream molecules of *Gdf5* might shed light on the physiology and pathology of articular chondrocytes. Epigenetic mechanisms such as DNA methylation, histone modifications, and noncoding RNAs regulate *Gdf5* expression. Moreover, identification of the transcription factors responsible for regulating the expression of *Prg4* and *Gdf5* will facilitate understanding of the molecular pathogenesis of OA and may provide causative therapies.

## Figures and Tables

**Figure 1 ijms-23-04672-f001:**
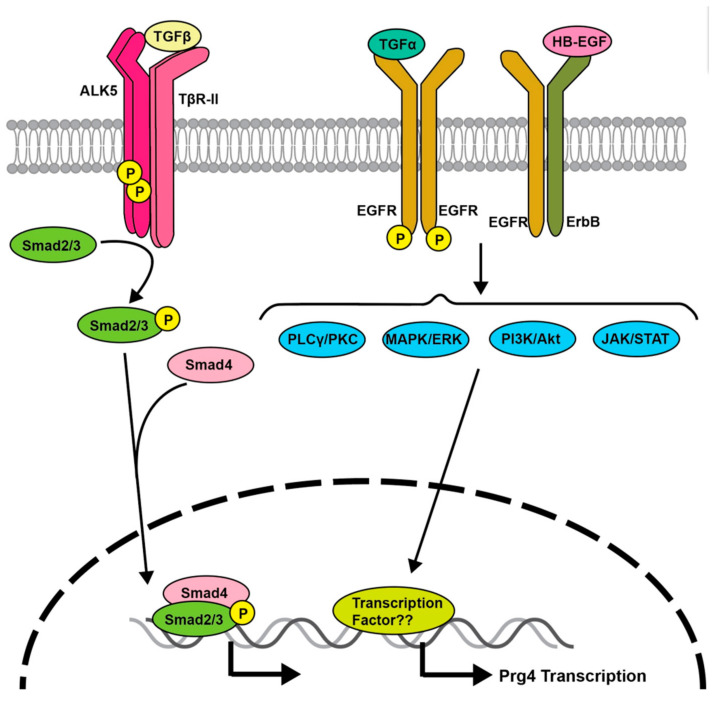
Schematic model of the TGF-β and EGFR signaling pathway in articular cartilage. Upon ligand binding, a heteromeric complex of type I (ALK5) and type 2 (TβR-II) receptors phosphorylate Smad2 and Smad3 (Smad2/3) effector proteins. Subsequently, phosphorylated Smad2/3 bind to Smad4 and translocate into the nucleus, where they bind to specific DNA sites to promote *Prg4* transcription. TGF-α binds to hetero- or homodimeric receptors of EGFR family members, consequently activating signals such as phospholipase C (PLCγ)/protein kinase C (PKC), MAPK/ERK, phosphatidylinositol-3 kinase (PI3K)/Akt, and JAK/signal transducer and activator of transcription (STAT) to promote *Prg4* transcription.

**Figure 2 ijms-23-04672-f002:**
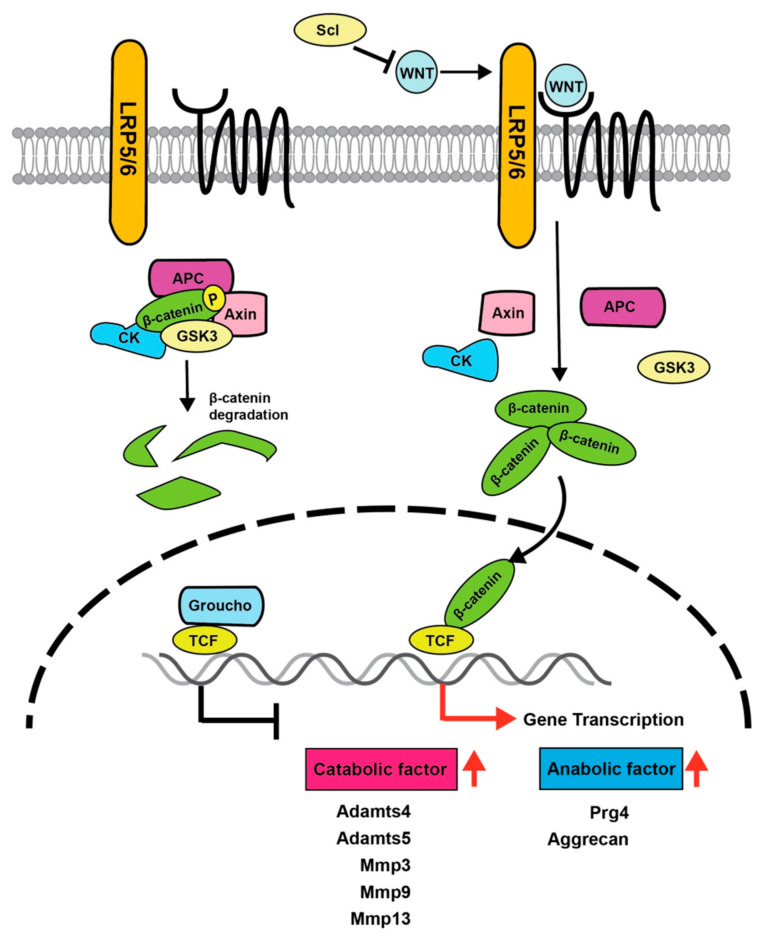
Schematic model of the canonical Wnt signaling pathway in the superficial zone of articular cartilage. Binding of WNTs to Frizzled and LRP5/6 co-receptors prevents complex formation with β-catenin, axin, adenomatous polyposis coli (APC), GSK, and CK, consequently causing β-catenin stabilization. Subsequently, β-catenin translocates from the cytoplasm into the nucleus, whereby it binds to TCF/LEF and promotes transcription. The genes induced include both catabolic factors for joint destruction and anabolic factors for joint protection. Accordingly, Wnt signaling contributes to the maintenance of articular cartilage homeostasis by balancing expression of these genes.

**Figure 3 ijms-23-04672-f003:**
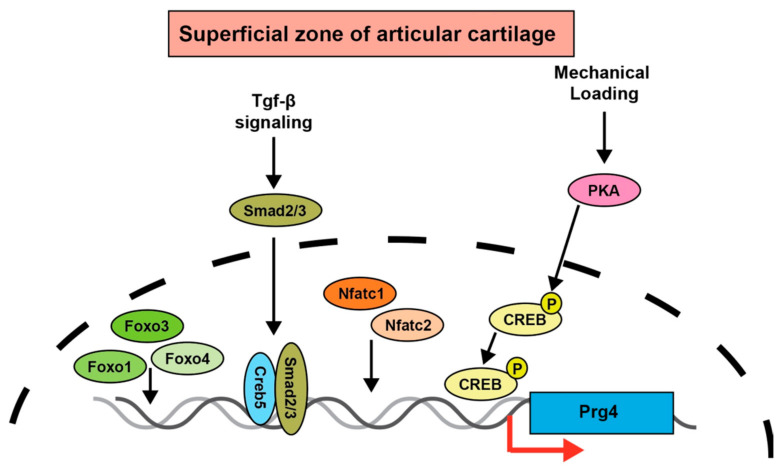
Schematic model of upstream transcription factors of *Prg4*. On the basis of current knowledge, the transcription factors listed in the figure seem to act directly upstream of *Prg4*. FoxO1, FoxO3 and FoxO4 induce *Prg4* expression in the superficial zone of articular cartilage. Creb5 acts cooperatively with Smad3 to induce *Prg4* expression. Mechanical stress activates protein kinase A (PKA) and induces *Prg4* expression via phosphorylation of CREB. Nfatc1 and Nfatc2 are transcription factors involved in induction of *Prg4* expression.

**Table 1 ijms-23-04672-t001:** Regulation factors of *Prg4* expression.

	*Prg4*
Upregulation factor	Wnt signal
	EGFR signal
	Tgf-β signal
	Foxo1/3/4 transcription factor
	Nfatc1, Nfatc2 transcription factor
	Creb 5
	Mechanical loading

**Table 2 ijms-23-04672-t002:** Regulation factors of Gdf5 expression.

	Gdf5
Upregulation factor	DMM model Pathological condition of OA
Down regulation factor	DNA methylation microRNA21-5p Yap

## Data Availability

Not applicable.

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
