# Peer review of "Regulatory Mechanisms of Prg4 and Gdf5 Expression in Articular Cartilage and Functions in Osteoarthritis"

_ijms, 2022, doi:10.3390/ijms23094672_

Round 1

Reviewer 1 Report

Manuscript deals with Prg4 and Gdf5 expression in articular cartilage and functions in OA. It is well organized, documented and written. Language is good. Conclusion is enough. However ıt has no material and method section. Authors should add this section which should include, data source, inclusion-exclusion criteria etc. Please see examples. My recommendation is minor revision.

Author Response

We greatly thank you for reviewing our manuscript and providing us with valuable comments.

We have prepared the manuscript following the IJMS Review guide. The guide is quoted below.

‘’Reviews offer a comprehensive analysis of the extant literature within a field of study, identifying current gaps or problems. They should be critical and constructive and provide recommendations for future research. No new, unpublished data should be presented. The structure can include an Abstract, Keywords, Introduction, Relevant Sections, Discussion, Conclusions, and Future Directions, with a suggested minimum word count of 4000 words.’’

According to it, the MATERIAL AND METHOD section is not included. Since this manuscript is a comprehensive analysis of the existing literature, we did not include a corresponding section on material and method.

Reviewer 2 Report

"Regulatory mechanisms of Prg4 and Gdf5 expression in articular cartilage and functions in osteoarthritis" by Takahata and colleagues is an interesting review of the literature that emphasises the role of proteoglycan (Prg4)/lubrin as a protective factor against osteoarthritis, as well as describing the regulatory mechanisms and functions of the differentiation factor Gdf5 in joint formation and the pathogenesis of osteoarthritis.
Overall, the manuscript is well structured and well organised. The figures are also well set up and explanatory, although they should be provided at a higher resolution.
Although the manuscript is full of interesting information, I suggest that the authors implement the concluding paragraph by discussing the reported information more critically. In addition, the authors could provide some more information about the role of Prg4 and Gdf5 as new strategies for the diagnosis and treatment of osteoarthritis. 
Finally, the manuscript needs a revision of the English. 

Author Response

We greatly thank you for reviewing and favorable evaluation of our manuscript and providing us with constructive comment.

1. The figures are also well set up and explanatory, although they should be provided at a higher resolution.

Reply: It is possible that the file resolution was lowered when it was converted to a PDF file on the peer review system. Our original figure maintains a high resolution and we will submit it.

2. Although the manuscript is full of interesting information, I suggest that the authors implement the concluding paragraph by discussing the reported information more critically. In addition, the authors could provide some more information about the role of Prg4 and Gdf5 as new strategies for the diagnosis and treatment of osteoarthritis.

Reply: We believe that Prg4 and Gdf5 play an important role in the maintenance of joint tissue homeostasis and may therefore be molecules that can serve as therapeutic strategies against osteoarthritis. Accordingly, in the revised manuscript, we have described the roles of Prg4 and Gdf5 above in the Conclusion section.

3. Finally, the manuscript needs a revision of the English.

Reply: Our manuscripts are checked by native English speakers. We have indicated in the Acknowledgements section that the manuscript was edited by Edanz group.

Reviewer 3 Report

The systematic review by Takahata et al., summarize findings about the role of proteoglycan 4 (Prg4)/lubricin, as a protective factor against Osteoarthritis (OA). They also focused on growth differentiation factor 5 (Gdf5), describing its regulatory mechanisms and functions in joint formation and OA pathogenesis. Given recently data on these genes in the maintenance of articular cartilage homeostasis of OA, molecular targeting of Prg4 and Gdf5 is expected to provide new insights into the aetiology, pathogenesis and potential treatment. The manuscript collects and analyzes interesting information to increase the knowledge of the regulatory mechanisms of Prg4 and Gdf5 expression in articular cartilage and functions in OA. The guidelines of International Journal of Molecular Sciences have been respected. So, it would be very useful for biomedical and biochemical researcher, and the scientific community in general. Therefore, it could be Accept with Minor Revision described below:
1. I recommend improving the English language to make it more homogeneous and making some minor changes to the manuscript, including the construction of shorter periods. Some sentences are long and complex, sometimes not very comprehensible.
2. In Figures 1, 2 and 3 the lettering is blurred. I recommend using higher resolution programs. If possible, enlarge the figures by inserting more descriptive details.
3. The manuscript lacks summary tables of the cited article, which are extremely useful to the reader in order to have a global and summary picture of the literature available at the moment. It would be useful to include two tables, one concerning the altered expression of the factor Prg4 and one concerning the factor Gdf5.
4. All gene names should be written in italics.
5. The conclusion should be implemented by better discussing the data presented and exploring the role these two factors could play in modulating new therapeutic strategies and approaches for diagnosing and monitoring OA.
6. It would be most interesting, based on the data available in the literature, to add a paragraph on the role of epigenetic mechanisms, such as DNA methylation, histone modifications and non-coding RNAs, in modulating the expression of these two factors.
7. In abstract section changes “etiology” with “aetiology.

Author Response

We greatly thank you for reviewing and favorable evaluation of our manuscript and providing us with constructive comment.

  1. I recommend improving the English language to make it more homogeneous and making some minor changes to the manuscript, including the construction of shorter periods. Some sentences are long and complex, sometimes not very comprehensible.

Reply:  Our manuscripts are checked by native English speakers. We have indicated in the Acknowledgements section that the manuscript was edited by Edanz group.

  1. In Figures 1, 2 and 3 the lettering is blurred. I recommend using higher resolution programs. If possible, enlarge the figures by inserting more descriptive details.

Reply:  It is possible that the file resolution was lowered when it was converted to a PDF file on the peer review system. Our original figure maintains a high resolution and we will submit it.

  1. The manuscript lacks summary tables of the cited article, which are extremely useful to the reader in order to have a global and summary picture of the literature available at the moment. It would be useful to include two tables, one concerning the altered expression of the factor Prg4 and one concerning the factor Gdf5.

Reply: Thank you for your very constructive comments.

We have attached a Summary table on Prg4 and Gdf5 control at the end of the revised manuscript.

  1. All gene names should be written in italics.

Reply: In the revised manuscript, all gene names are in italics, and those that refer to proteins are in the standard form.

  1. The conclusion should be implemented by better discussing the data presented and exploring the role these two factors could play in modulating new therapeutic strategies and approaches for diagnosing and monitoring OA.

Reply: We believe that Gdf5 and Prg4 are important for maintaining joint tissue homeostasis. The details of the function of these genes during the pathogenesis of osteoarthritis are currently under analysis and will be addressed in the future. In the revised manuscript, we added a paragraph with possibility of two factors on osteoarthritis monitoring in conclusion section.

  1. It would be most interesting, based on the data available in the literature, to add a paragraph on the role of epigenetic mechanisms, such as DNA methylation, histone modifications and non-coding RNAs, in modulating the expression of these two factors.

Reply: We greatlty thank you for valuable comment. We added a paragraph on the role of epigenetic mechanisms, in modulating Gdf5 expression.

  1. In abstract section changes “etiology” with “aetiology.

Reply: As pointed by the reviewer, we have replaced ‘etiology’ with ‘aetiology’ in the abstract.

This manuscript is a resubmission of an earlier submission. The following is a list of the peer review reports and author responses from that submission.